# Domain-Aware Adaptive Logarithmic Transformation

Xuelai Fang [†]  and Xiangchu Feng *,[†]

School of Mathematics and Statistics, Xidian University, Xi'an 710071, China
* Correspondence: xcfeng@mail.xidian.edu.cn
† These authors contributed equally to this work.

**Abstract:** Tone mapping (TM) aims to display high dynamic range scenes on media with limited visual information reproduction. Logarithmic transformation is a widely used preprocessing method in TM algorithms. However, the conventional logarithmic transformation does not take the difference in image properties into account, nor does it consider tone mapping algorithms, which are designed based on the luminance or gradient-domain features. There will be problems such as oversaturation and loss of details. Based on the analysis of existing preprocessing methods, this paper proposes a domain-aware adaptive logarithmic transformation AdaLogT as a preprocessing method for TM algorithms. We introduce the parameter $p$ and construct different objective functions for different domains TM algorithms to determine the optimal parameter values adaptively. Specifically, for luminance-domain algorithms, we use image exposure and histogram features to construct objective function; while for gradient-domain algorithms, we introduce texture-aware exponential mean local variance (EMLV) to build objective function. Finally, we propose a joint domain-aware logarithmic preprocessing method for deep-neural-network-based TM algorithms. The experimental results show that the novel preprocessing method AdaLogT endows each domain algorithm with wider scene adaptability and improves the performance in terms of visual effects and objective evaluations, the subjective and objective index scores of the tone mapping quality index improved by 6.04% and 5.90% on average for the algorithms.

**Keywords:** high dynamic range; tone mapping; adaptive logarithmic transformation; preprocessing



## 1. Introduction

The dynamic range of images is defined as the logarithm of the ratio of the maximum to the minimum luminance [1]. Through high dynamic range (HDR) images, we can restore the human eye's perception of scenes as much as possible [2]. Since the dynamic range of natural scenes often exceeds the display range of low dynamic range (LDR) images, traditional display devices cannot directly display HDR images well. Therefore, how to map HDR images to traditional displays and show them well has become one of the current research hotspots in image processing.

Tone mapping (TM) compresses the dynamic range of an image, mapping high contrast, wide gamut HDR images onto conventional display devices. In general, tone mapping algorithms consist of two parts: preprocessing and tone mapping. The pipeline is shown in Figure 1.

The TM algorithms' intentions can be classified as scene reproduction, best subjective quality and visual system simulator [3]. For scene reproduction, a variety of data processing methods are used in tone mapping operators (TMO), among which the logarithmic transformation is a simple and widely used one. Studies have shown that TM algorithms are closely related to human visual system (HVS) perception. The Weber–Fechner law [4] shows the sensitivity of HVS to luminance variations: the response of HVS in most of the luminance range has logarithmic characteristics, namely, there is a logarithmic relationship between the perceived luminance and physical luminance [3]. Thus, many TM algorithms

choose to compute in the logarithmic domain to ensure consistency between perceived luminance and scene luminance.

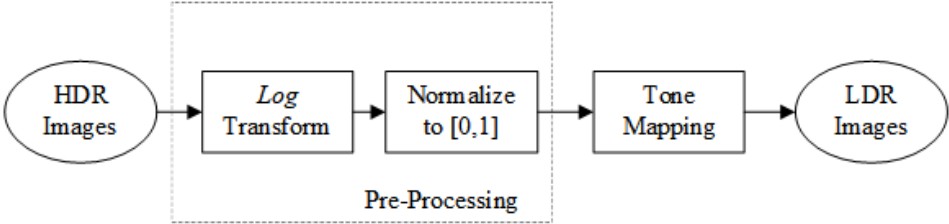

**Figure 1.** The pipeline of TM algorithms.

The TM algorithms process images from different domains. Traditional TM algorithms can be divided into luminance-domain methods and gradient-domain methods [2]. The layer decomposition method is representative of the luminance-domain algorithm. Durand et al. [5] proposed a single-scale decomposition of HDR images in the luminance domain using bilateral filter instead of Gaussian filter. Farbman et al. [6] constructed an edge-preserving filter based on the weighted least square (WLS) method and used multi-scale decomposition to further enhance the discrimination of low/high-frequency information in the image. The results show that the WLS algorithm suppresses the halo problem of Durand's method well. Paris et al. [7] proposed a tone mapping method based on the image Laplace pyramid with better halo reduction and detail retention properties. Liang et al. [8] introduced $l_0, l_1$ priors for different layers in the image decomposition process. Yang et al. [9] adaptively selected two appropriate gamma functions to adjust the brightness of dark and light areas, respectively. However, the dynamic range compression may lead to a little loss of visual naturalness. Beyond that, Dargo [10] first proposed an adaptive logarithmic tone mapping curve. Zhao et al. [11,12] proposed the effective TMOs by using localized contrast correction and Retinex [13]. Mantiuk et al. [14] used the contrast perturbation of the HVS model as the weight to construct the tone mapping operator, and Khan et al. [15] adjusted the image luminance histogram based on the just noticeable difference (JND) and used a look-up table to construct the mapping.

On the other hand, the gradient-domain algorithm performs dynamic range compression and detail enhancement by manipulating image gradients. Fattal et al. [16] constructed a compression function using multi-scale Gaussian pyramid to compress the large gradient of the image while keeping the small gradient unchanged or enhanced, which has the advantages of detail preservation and almost no halo effect. Bhat et al. [17] proposed a unified framework for gradient-domain image processing. Shibata et al. [18] combined the gradient-domain algorithm with the luminance-domain algorithm to avoid oversaturation and gradient reversal using luminance constraints. In addition, deep neural network (DNN)-based algorithms [19–22] have also been emerging in recent years, achieving significant advantages.

Many of the above algorithms use conventional logarithmic transformation LogT or its variants for preprocessing, without considering the diversity of natural scenes and various luminance ranges of different scenes. These preprocessing methods are also not tuned for the different domains TM algorithms, resulting in problems such as oversaturation and loss of details in the mapped images. Based on the analysis of various existing preprocessing methods, this paper proposes a domain-aware adaptive logarithmic transformation AdaLogT as a unified TM preprocessing method. We introduce the parameter $p$ and construct different objective functions for luminance and gradient domains to determine the optimal parameter value of $p$. Specifically, for luminance-domain algorithms, we use image exposure and histogram features to construct the objective function to maximize the layered performance of the luminance-domain algorithms. For gradient-domain algorithms, texture-aware exponential mean local variance (EMLV) [23] is introduced to build the objective function to ensure the maximization of the input gradient information. Based on these, we propose a joint domain-aware logarithmic preprocessing method for DNN-based TM algorithms. The experimental results show that the proposed preprocessing

method endows each domain algorithm with wider scene adaptability and improves the performance in terms of visual effects and objective evaluations.

The rest of this paper is organized as follows. Section 2 analyzes the related preprocessing algorithms and proposes an adaptive logarithmic transformation model AdaLogT. Section 3 describes the objective functions corresponding to the luminance-domain and gradient-domain TM methods. Section 4 proposes a joint domain-aware logarithmic transformation for the DNN-based TM methods. Then, Section 5 presents the experimental results of subjective and objective comparisons with existing methods. Finally, Section 6 concludes this paper and outlooks for further work.

## 2. Related Work and Adaptive Logarithm Transformation Model

The luminance range of HDR images is approximately 0.0005 cd/m$^2$ to 10,000 cd/m$^2$ [24]. It is necessary to normalize the preprocessing so that the image pixel values fall within a specific range, reducing the computational complexity and ensuring the effectiveness of TM algorithms. In this section, we first review classic preprocessing methods. The summary of these research is shown in Table 1.

Let the input image be $I$, and image $\bar{I}$ in logarithmic transformation can be expressed as:

$$\bar{I} = log(I + \epsilon) \tag{1}$$

where $\epsilon = 1 \times 10^{-4}$. Considering the difference in dynamic range of different images, the normalization as follows is used on (1):

$$\widetilde{I} = \frac{\bar{I} - \bar{I}_{min}}{\bar{I}_{max} - \bar{I}_{min}} \tag{2}$$

where $\bar{I}_{min}$ and $\bar{I}_{max}$ represent the minimum and maximum pixel values of $\bar{I}$, respectively. $\widetilde{I}$ indicates the image after logarithmic transformation and normalization. Then, the range of pixel values are normalized to $[0, 1]$. This method is simple enough and has a good display effect. Liang [8] and other works [16,25,26] use it as a preprocessing step in the TM algorithms. We denote this method as the traditional logarithmic transformation LogT.

Stockham [27] recommends that the image should satisfy the following logarithmic relationship for image processing and display needs:

$$\widetilde{I} = \frac{log(I + 1)}{log(I_{max} + 1)} \tag{3}$$

Dargo [10] proposed an adaptive logarithmic tone curve for tone mapping. A bias power function is introduced to adaptively vary logarithmic bases. The algorithm changes the mapping of scene brightness and contrast by different logarithmic bases. Equation (4) present the tone mapping function:

$$\widetilde{I} = \frac{I_{dmax} \cdot 0.01}{log_{10}(I_{max} + 1)} \frac{log(I + 1)}{log(2 + ((\frac{I}{I_{max}})^{\frac{log(b)}{log(0.5)}} \cdot 8)} \tag{4}$$

where $I_{dmax}$ is used as a scalefactor to adapt the output to its intended display. Generally, the reference value of $I_{dmax}$ for displays is set at 100 cd/m$^2$. Adjusting the bias function parameter $b$ is equivalent to adjusting the base of the logarithmic function, thus changing the overall effect of the result.

Gu [28] found that appealing results could be obtained by appropriately amplifying the input luminance. According to the dynamic range of conventional scenes, the following logarithmic transformation and normalization are given:

$$\widetilde{I} = \frac{log(I \cdot 10^6 + 1)}{log(I_{max} \cdot 10^6 + 1)}, \tag{5}$$

Recently, Vinker [21] proposed adaptive curve-based compression (ACC) for preprocessing of DNN algorithms, which maps and normalizes the input image through the following transformations:

$$\widetilde{I} = \frac{log(\lambda \cdot \frac{I}{I_{max}} + \varepsilon)}{log(\lambda + \varepsilon)}, \tag{6}$$

where $\lambda$ is the scaling factor and the selection rule of $\lambda$ is to minimize the following cross-entropy:

$$\arg\min_{\lambda} - \sum_{l} H_l(\widetilde{I}) log(H_l(LDR)), \tag{7}$$

where $H(\cdot)$ represents the histogram. $H(\widetilde{I})$ as a function of $\lambda$ denotes the histogram of $\widetilde{I}$, and $H(LDR)$ represents the histogram of native LDR images. $H(LDR)$ is obtained by averaging the histogram of 900 high-quality images in the DIV2k [29] dataset. All histograms use 20 bins indexed by $l$.

Inspired by the above works, we construct a unified TM algorithms' preprocessing format named adaptive logarithmic transformation AdaLogT:

$$\widetilde{I} = \frac{log(I \cdot 10^p + 1)}{log(I_{max} \cdot 10^p + 1)} \triangleq AdaLogT(I; p), \tag{8}$$

where $I_{max}$ represents the maximum value of image $I$. $\widetilde{I}$ is strictly limited to the range $[0, 1]$ after normalization.

Equations (2), (3), and (5) can be expressed in the form of Equation (8). In fact, Equation (5) corresponds to the special case of $p = 6$ in Equation (8). For Equation (2), we have

$$\widetilde{I} = log(I + \epsilon) = log(\frac{1}{\epsilon} \cdot I + 1) + log(\epsilon), \tag{9}$$

For $\widetilde{I}_{min}, \widetilde{I}_{max}$, there are

$$\widetilde{I}_{min} = log(I_{min} + \epsilon) = log(\frac{1}{\epsilon} \cdot \widetilde{I}_{min} + 1) + log(\epsilon), \tag{10}$$

$$\widetilde{I}_{max} = log(I_{max} + \epsilon) = log(\frac{1}{\epsilon} \cdot \widetilde{I}_{max} + 1) + log(\epsilon), \tag{11}$$

Since $\widetilde{I}_{min} = 0$, bringing Equations (9)–(11) into Equation (8) has

$$\begin{aligned}
\widetilde{I} &= \frac{log(\frac{1}{\epsilon} \cdot I + 1) - log(\frac{1}{\epsilon} \cdot I_{min} + 1)}{log(\frac{1}{\epsilon} \cdot I_{max} + 1) - log(\frac{1}{\epsilon} \cdot I_{min} + 1)} \\
&= \frac{log(\frac{1}{\epsilon} \cdot I + 1)}{log(\frac{1}{\epsilon} \cdot I_{max} + 1)}, (0 \leqslant \widetilde{I} \leqslant 1)
\end{aligned} \tag{12}$$

Therefore, Equation (8) is a generalized form of Equations (2) and (5). The parameter $p$ enables us to adaptively obtain a suitable log-normalized transformation for different input images, which represents the order of magnitude of image amplification as shown in Figure 2. Using the parameter $p$ to amplify the input luminance is equivalent to performing the corresponding global compression mapping, which has the properties of enhancing the contrast of low-luminance while compressing the dynamic range of high-luminance for all pixels of the image.

With the increase in the parameter $p$, the range of enhancement area reduced, and the amplitude of enhancement increased. Meanwhile, the suppression effect of the highlighted area is enhanced, and the overall luminance of the image is improved. When $p$ is too large, the dynamic range of the original low-luminance region is also compressed, and the image details will be suppressed, resulting in the lack of contrast. For HDR images where most of the data is located in the low-luminance region and a small part of the data have the characteristics of very high luminance, the selection of parameter $p$ is essentially a

trade-off between the detail enhancement region and the dynamic range compression region. From Figure 2, we can select the range of $p$ as $[-5, 10]$. For different input images, the parameter $p$ is selected adaptively according to image information and the domain of the TM algorithm. Specific selection strategies will be given in the following sections.

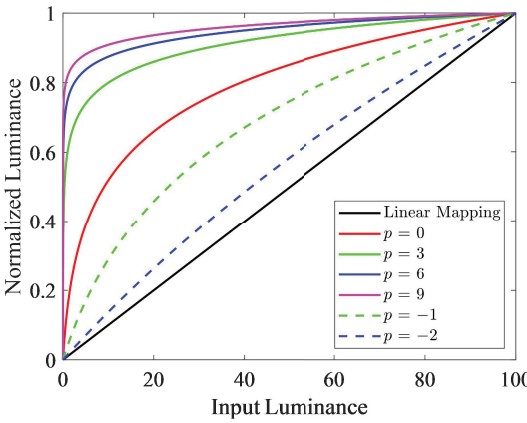

**Figure 2.** Mapping curves with different $p$ values.

**Table 1.** Summary of tone mapping preprocessing research.

| Researcher | Expression | Advantage | Disadvantage |
|---|---|---|---|
| Stockham [27] | $\frac{log(I+1)}{log(I_{max}+1)}$ | Strictly mapped to the interval $[0, 1]$ | The luminance compression is excessive; The lost of high contrast content. |
| Dargo [10] | $\frac{I_{dmax}\cdot 0.01}{log_{10}(I_{max}+1)}\frac{log(I+1)}{log(2+((\frac{I}{I_{max}})^{\frac{log(b)}{log(0.5)}}\cdot 8)}$ | Well-suited to the specific image content | Parameter $b$ needs to be adjusted for different images; Local contrast reduction |
| Gu [28] | $\frac{log(I\cdot 10^6+1)}{log(I_{max}\cdot 10^6+1)}$ | Enhancing low-light areas of the image; Improving the overall brightness of the image | Overexposure may occur |
| Vinker [21] | $\frac{log(\lambda\cdot\frac{I}{I_{max}}+\varepsilon)}{log(\lambda+\varepsilon)}$ | Adaptive searching for appropriate mapping curves | High computational complexity; Not strictly normalized to the interval $[0, 1]$ |

## 3. Domain-Aware Objective Function

Traditional TM algorithms can be classified into luminance-domain and gradient-domain methods broadly. The luminance-domain methods use layer decomposition, histogram [15,30,31], HVS [32–34], etc. to deal with image luminance. These types of methods consider how to compress the HDR image luminance to the display range of traditional display devices. The gradient-domain methods focus on the preservation of image contrast and gradient, directly acting on the image gradient to achieve overall dynamic range compression. Based on their different focus directions, we propose different objective functions in the luminance domain and gradient domain to guide the selection of the parameter $p$, which are called luminance-domain-aware AdaLogT methods and gradient-domain-aware AdaLogT methods, respectively. The overall flowchart of the proposed method is shown in Figure 3.

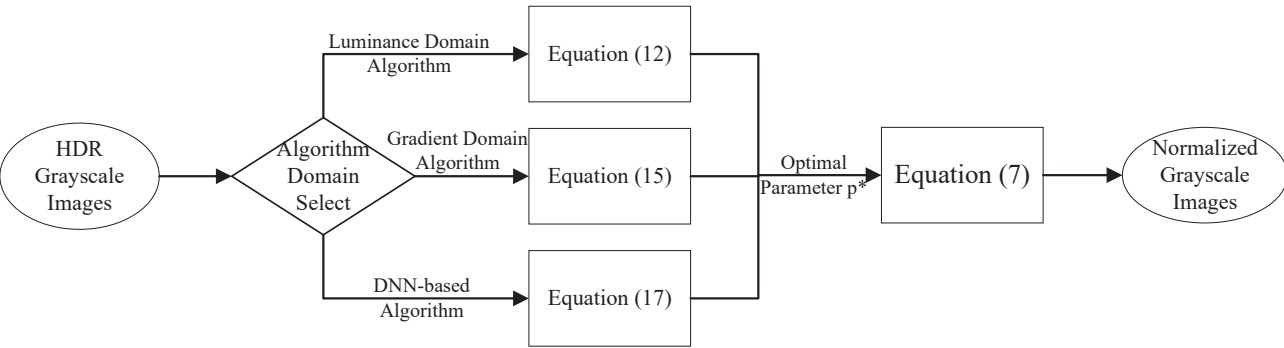

**Figure 3.** Overall flowchart of the proposed method.

### 3.1. Luminance-Domain-Aware AdaLogT Method

The grayscale mean of image pixels reflects the exposure degree of the image [35–37]. Direct observation shows that the distribution of HDR image pixels at each luminance level is uneven [38], as shown in Figure 4a. On the other hand, histogram equalization states that if the pixels of an image can be distributed evenly over all possible gray levels, the image will have high contrast and richer details. In AdaLogT, different parameters $p$ can adjust the luminance distribution of the image without changing the overall shape of the image histogram, thereby adjusting the exposure level of the image. Figure 4 shows the image grayscale histogram under different parameters. With the logarithmic transformation of different $p$ values, the image histogram gradually extends to other luminance levels.

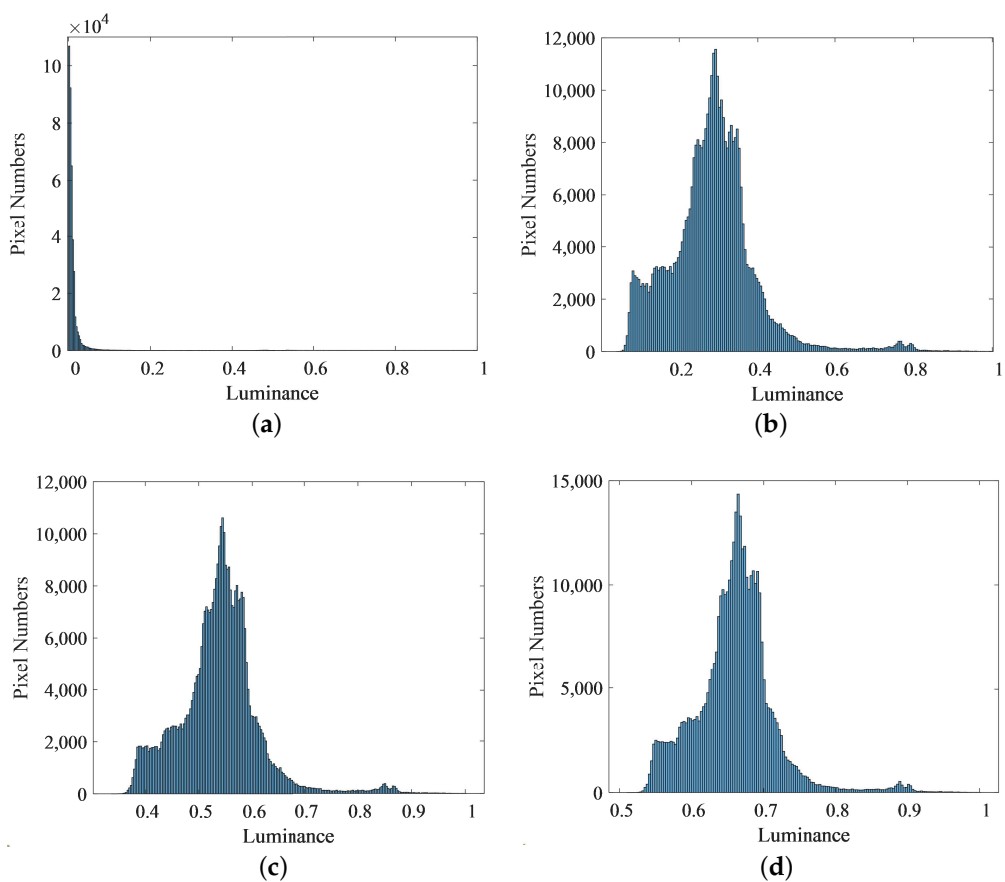

**Figure 4.** Image histogram distribution under different parameters $p$: (**a**) radiance map; (**b**) $p = 3$; (**c**) $p = 6$; (**d**) $p = 9$.

Image skewness $s(\cdot)$ measures the symmetry of the image distribution concerning the mean value, and the degree of its approximation to 0 reflects the degree of symmetry of the distribution. Therefore, we introduce image skewness to ensure the symmetry of the image distribution after adjustment. Thus, the objective function and corresponding optimization problem for luminance-domain TM algorithms are given as follows:

$$
\begin{aligned}
p^* &= \arg\min_{p} \alpha \cdot \left\| \widetilde{I} - T \right\|_F^2 + (1 - \alpha) \cdot \left| s(\widetilde{I}) \right|, \\
s.t. \ \widetilde{I} &= log(I \cdot 10^p + 1)/log(I_{max} \cdot 10^p + 1), \\
s(\widetilde{I}) &= \mathbb{E}(\widetilde{I} - \mu)^3/\sigma^3,
\end{aligned}
\tag{13}
$$

where $\|\cdot\|_F$ indicates Frobenius norm. $s(\widetilde{I})$ is the skewness of image $\widetilde{I}$, $|s(\widetilde{I})|$ means the absolute value of $s(\widetilde{I})$, and $\mathbb{E}(\cdot)$ denotes the expectation. $\mu$ and $\sigma$ are respectively the mean and variance of $\widetilde{I}$. $T$ represents the exposure level of the target image, and $\alpha$ is the weight to balance exposure and skewness. This paper defaults to $T = 0.5$ and $\alpha = 0.8$.

We can calculate the above optimization problem with the trichotomy method to obtain the optimal parameter $p$, as shown in Algorithm 1. Figure 5 shows the grayscale histograms of some images before and after adaptive logarithmic transformation. By comprehensively considering exposure and skewness, the AdaLogT images have better display luminance and contrast, which is reflected in the histogram, that is, the pixels are distributed to as many grayscale levels as possible.

---

**Algorithm 1:** Trichotomy method for optimum value

---

> **input　:** Grayscale high dynamic range image $I$, Exposure level $T$, Lower bound $l$,
> 　　　　　Upper bound $L$, weight $\alpha$.
> **output:** $p^*$

**1** $f(I) := \alpha \cdot \left\| I - T \right\|_F^2 + (1 - \alpha) \cdot \left| s(I) \right|$;

**2** $I_p := log(I \cdot 10^p + 1)/log(I_{max} \cdot 10^p + 1)$;

**3 while** $l <= L$ **do**

**4**　　$l_{mid} = (l + L) >> 1$, $L_{mid} = (l_{mid} + L) >> 1$;

**5**　　Compute $I_{l_{mid}}, I_{L_{mid}}, f(I_{l_{mid}}), f(I_{L_{mid}})$;

**6**　　**if** $f(I_{l_{mid}}) <= f(I_{L_{mid}})$ **then**

**7**　　　$\lfloor$　$L = L_{mid} - 1$

**8**　　**else**

**9**　　　$\lfloor$　$l = l_{mid} + 1$

**10** $p^* = l$

---

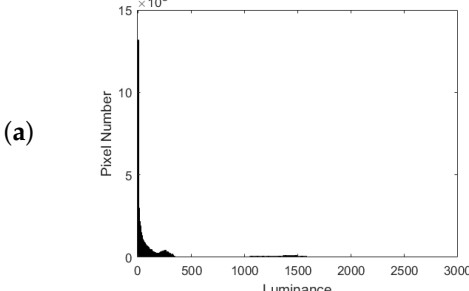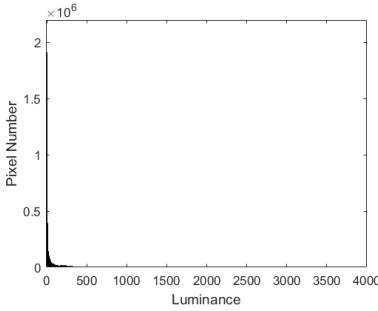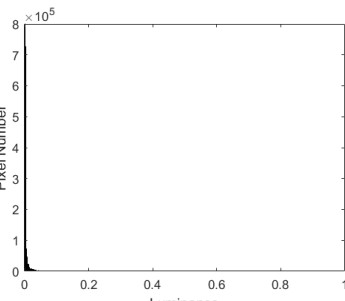

**(a)**

**Figure 5.** *Cont.*

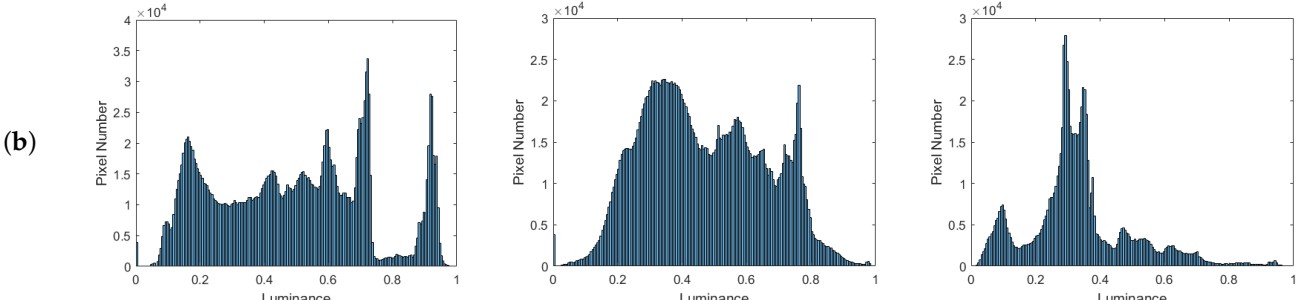

**Figure 5.** Histogram before and after adaptive logarithmic transformation of different images: (**a**) radiance map; (**b**) AdaLogT.

### 3.2. Gradient-Domain-Aware AdaLogT Method

For gradient-domain algorithms, the enhancement of image details by preprocessing is the most critical. The gradient-domain algorithm represented by [16] reconstructs the image by solving the Poisson equation:

$$\Delta f = div(G) \tag{14}$$

where $\Delta$ is the Laplace operator. $div(\cdot)$ denotes the divergence. $f$ is the output image to be reconstructed, and $G$ is the guided gradient field calculated according to the log-transformed image gradients. The guided gradient field maintains the order relationship of the original image gradients and compresses the large gradients while enhancing the small gradients. However, the enhancement of small gradients depends on the input image gradients, which leads to the problem that the reconstructed image is too dark to distinguish details due to the small gradients of the input image detail part.

Based on this observation, the key to selecting the logarithmic transformation parameters of gradient-domain algorithms is to ensure the image has good detail performance. Therefore, the mean of exponential mean local variance (EMLV) [23] is introduced as the measure of image detail, denoted as $M_g$:

$$M_g = \frac{1}{N} \sum_{i=1}^{N} \left| \frac{1}{|\Omega|} \sum_{\Omega} \nabla \widetilde{I}_i \right|^\gamma, \tag{15}$$

where $\widetilde{I}_i$ represents the *ith* pixel of image $\widetilde{I}$, and $\Omega$ is a $3 \times 3$ neighborhood of $\widetilde{I}_i$. $|\Omega|$ means the number of pixels in $\Omega$, and $N$ denotes the total number of pixels in $\widetilde{I}$. $|\nabla \widetilde{I}_i| = \sqrt{(\partial_x \widetilde{I}_i)^2 + (\partial_y \widetilde{I}_i)^2}$, $\partial_i$ denotes the partial derivative with respect to the direction $i$. $\gamma$ determines the sensitivity to the gradient of $\widetilde{I}$, and $\gamma$ is taken as 0.5 in this paper.

Figure 6 shows the change of $M_g$ after changing the parameter $p$ in different images. As the parameter $p$ increases, the $M_g$ value of the log-transformed image presents a single peak that first increases and then decreases. In further experiments, when $p$ is small, the image luminance is low, and texture details are lost. While the image luminance and contrast decrease when $p$ is too high. Therefore, we suggest that the parameter $p$ be selected to maximize $M_g$ after transformation to ensure the maximization of input gradient information. We give the optimization problem corresponding to the objective function of the gradient-domain TM algorithms:

$$p^* = \underset{p}{\arg\max} \frac{1}{N} \sum_{i=1}^{N} \left| \frac{1}{|\Omega|} \sum_{\Omega} \nabla \widetilde{I}_i \right|^\gamma, \tag{16}$$

$$s.t. \ \widetilde{I} = log(I \cdot 10^p + 1)/log(I_{max} \cdot 10^p + 1),$$

Due to the unimodality of the objective function, we can use a zero-order optimization method such as the Fibonacci method to calculate the optimal value of $p$.

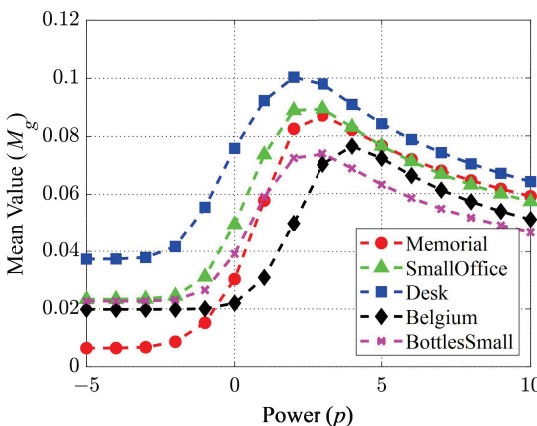

**Figure 6.** The relationship between the mean value of $M_g$ and $p$.

## 4. *AdaLogT* Method for DNN-Based TM Algorithms

Many algorithms, such as the DNN-based TM algorithm, do not solely consider image luminance or gradient information. DNN algorithms are data-driven and learn the main features of TM process through a large number of samples, which include but are not limited to image luminance and gradient, etc. Some of its convolution operations may contain the function of the average operator, while others may contain the function of the difference operator. Therefore, the objective functions for a single domain may not enhance all the information required by the algorithm.

In Vinker [21] , a log-normalized preprocessing method for DNN-based TM algorithms was proposed. For Equations (6) and (7), the following two issues need further study and discussion.

(1)   Normalization. $\widetilde{I}_{max} = 1$, but $\widetilde{I}_{min} = \frac{log(\epsilon)}{log(\lambda+\epsilon)} \neq 0$ when $I_{min} = 0$. In other words, Equation (6) does not strictly map the input luminance to $[0, 1]$. If we modify Equation (6) to:

$$\widetilde{I} = \frac{log(\lambda \cdot \frac{I}{I_{max}} + \varepsilon) - log(\epsilon)}{log(\lambda + \varepsilon) - log(\epsilon)}, \tag{17}$$

Then $\widetilde{I} \in [0, 1]$.In this case $\frac{\lambda}{\epsilon \cdot I_{max}} = 10^p$, the selection of $\lambda$ is transformed into the problem of selection of $p$.

(2)   Computational complexity. Equation (7) uses the mean of the luminance histograms of 900 LDR images in the DIV2k [29] dataset as reference. Ideally, the calculation of the histogram means should use the distance between distributions, such as earth mover's distance (EMD) [39], which is computationally expensive. Specifically, Vinker uses the stochastic search method [40] to find suitable values within 1 to $1 \times 10^9$ and uses a floating point type with a high degree of computational accuracy, which needs to be continually performed. Depending on the variation of the mapping curve with different parameters in Figure 2, there is less gain in increased accuracy as it takes a large parameter change to make a significant difference to the curve. Figure 7 gives a comparison of ACC and AdaLogT execution times and shows that ACC has a far greater computational complexity than AdaLogT.

Using the analysis in Section 3, we know that since the DNN contains both the luminance domain and the gradient domain, the corresponding objective function should have the form of joint domain perception:

$$
p^* = \arg\min_{p} \alpha \cdot \left( \left\| \widetilde{I} - T \right\|_F^2 \right) + (1-\alpha) \cdot \left| s(\widetilde{I}) \right|
$$
$$
- \frac{1}{N} \sum_{i=1}^{N} \left| \frac{1}{|\Omega|} \sum_{\Omega} \nabla \widetilde{I}_i \right|^{\gamma},
$$
$$
s.t. \ \widetilde{I} = log(I \cdot 10^p + 1)/log(I_{max} \cdot 10^p + 1),
$$

(18)

Equation (18) integrates image luminance and gradient features with wide perceptual range, which can be solved by methods such as step-by-step method. Compared with Equation (7), the computational cost of Equation (18) is greatly reduced.

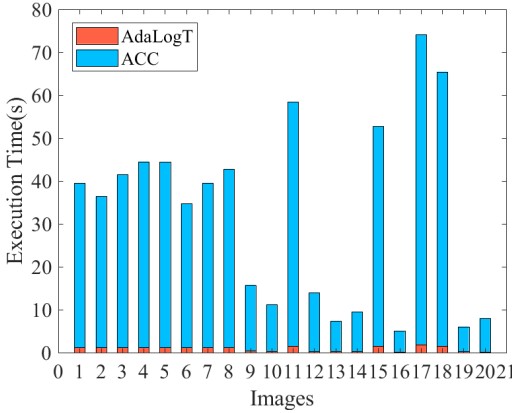

**Figure 7.** Comparison of ACC andAdaLogT image execution time.

## 5. Experimental Results and Analysis

The state-of-the-art TM algorithms used for comparison in the experiment are the luminance-domain algorithm Gu [28], the gradient-domain algorithm Fattal [16], and the deep neural network algorithm Vinker [21]. The source codes of Gu [28] and Vinker [21] are obtained from the authors' homepage, and we use the default parameters of the programs. Additionally, the network pre-training parameters given by Vinker [21] are used as the default. The Fattal [16] algorithm is implemented by 'LuminanceHDR ' (https://qtpfsgui.sourceforge.net/, accessed on 13 September 2022) and gamma is set to 2.2. All these experiments were run on a HP Workstation Z680 with Intel Xeon E5630 CPU, NIVDIA GeForce 2080Ti GPU and 32 GB memory. To fully consider the differences in different scenes, we perform experiments on a large number of HDR images and randomly select 20 images for experimental analysis.

### 5.1. Luminance-Domain Algorithm

The preprocessing method proposed by Gu [28] provides a better display for brighter scenes. However, for indoor and outdoor dark scenes, there will be problems where the background luminance of output images does not match the real scene, and the overall contrast is reduced. The luminance-aware AdaLogT better considers the background luminance of different scenes and enhances the subjective visual effect. Figure 8 shows the comparison of the two methods. A subjective experiment is conducted based on the results of 20 HDR images. We invite ten people to evaluate the experimental results, 6 males and 4 females, six of whom have a research area in image processing. The rating scale is from 1 (worst) to 10 (best) in steps of 1. The results are displayed on Samsung S32R750UEC 32 inch (4096 × 2160). Compared with the mean and standard deviation of Gu's method (6.84, 1.55), our method (7.40, 1.15) achieved an 8% improvement.

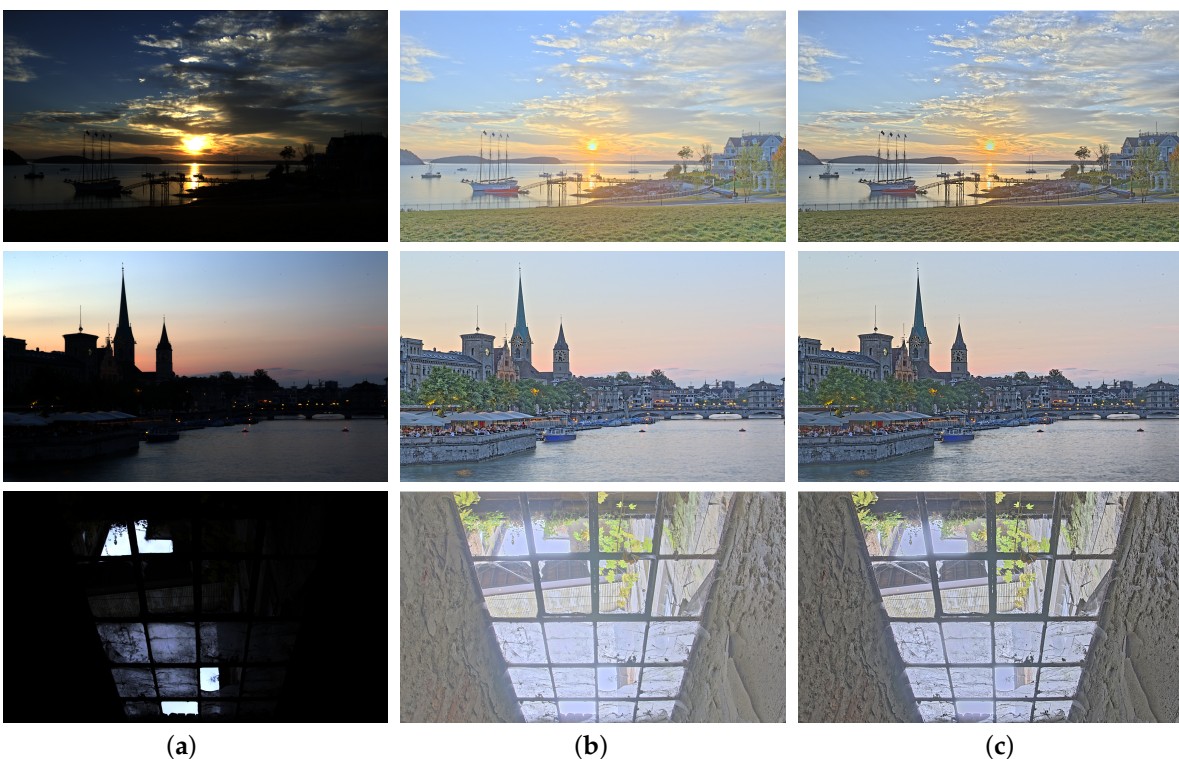

(**a**)　　　　　　　　　(**b**)　　　　　　　　　(**c**)

**Figure 8.** Comparison of the results of the luminance-domain TM algorithm in different preprocessing methods: (**a**) radiance map; (**b**) Gu [28]; (**c**) AdaLogT.

We also select the tone-mapped image quality index (TMQI) [41] for objective evaluation. TQMI evaluates images from multiple perspectives. This method measures the structural fidelity and naturalness scores of tone-mapped results. Then, it comprehensively gives a final score ranging from 0 to 1. A larger value of TMQI represents better result achieved by the TM algorithm. A scatter plot is used to visualize the TMQI scores of different preprocessing methods on experimental images. As shown in Figure 9, AdaLogT achieves better results in most images.

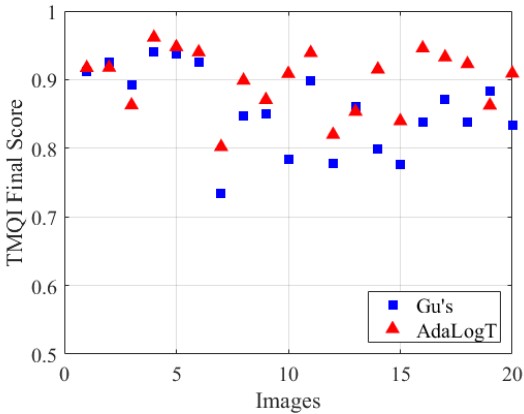

**Figure 9.** TMQI final score of the luminance-domain algorithm in different preprocessing methods.

Table 2 shows the mean TMQI scores of 20 experimental images before and after the adoption of AdaLogT, where the highest score is given in bold. We observe that AdaLogT improves the adaptability of the TM algorithm to different scenes. The appropriate exposure choice brings great advantages to the image display, effectively improving the naturalness score of TMQI and resulting in a higher TMQI final score.

**Table 2.** Mean TMQI scores of luminance-domain algorithm.

| Preprocessing | Structure | Naturalness | Final |
|---|---|---|---|
| Gu's | 0.8273 | 0.4098 | 0.8562 |
| AdaLogT | 0.8305 | 0.6346 | 0.8983 |

*5.2. Gradient-Domain Algorithm*

The gradient-domain algorithm [16] based on the gradient-domain-aware AdaLogT method has achieved good results in quantitative evaluation and objective assessment.

Figure 10 shows the comparison of TM results for bright and dark scenes, where the first column is the HDR radiance map, the second column is the TM results under LogT, and the third column is the TM results using AdaLogT. When the input image is dark, the result under LogT is dim and the details are vague or even indistinguishable. Our method corrects image exposure and achieves a balance between detail preservation and image naturalness. The structure and local details of the image are better preserved in the result, and the visual effect is more consistent with the human eye's perception of the scene. In the quantitative user evaluation, the Fattal method and ours achieve scores of (6.73, 1.13) and (7.16,1.07), respectively.

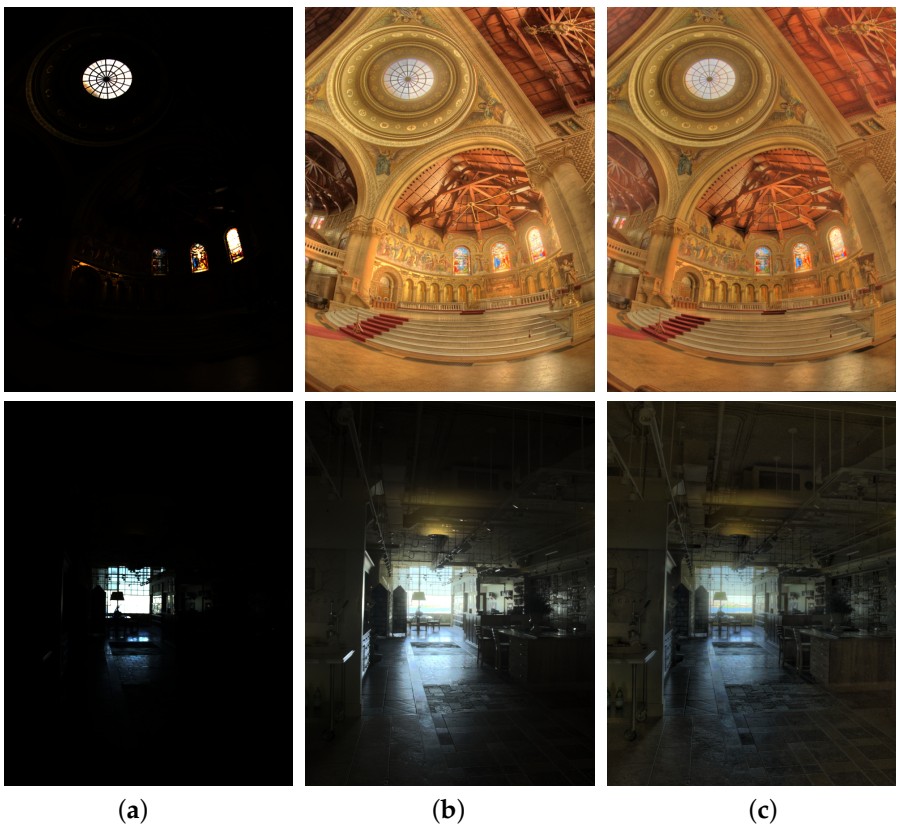

(**a**)          (**b**)          (**c**)

**Figure 10.** Comparison of the results of the gradient-domain TM algorithm in different preprocessing methods: (**a**) radiance map; (**b**) LogT; (**c**) AdaLogT.

Figure 11 is the scatter plot of the gradient-domain algorithm results under different preprocessing methods. Our algorithm achieves equal or better scores in the vast majority of images. Table 3 presents the mean TMQI scores under preprocessing methods LogT and AdaLogT. The gradient-domain objective function pays attention to image details and textures so that the TM results have better visual brightness and detail preservation. Therefore, compared with the result of LogT, the result of AdaLogT has a higher TMQI structure score, and has also achieved significant improvement in the naturalness index.

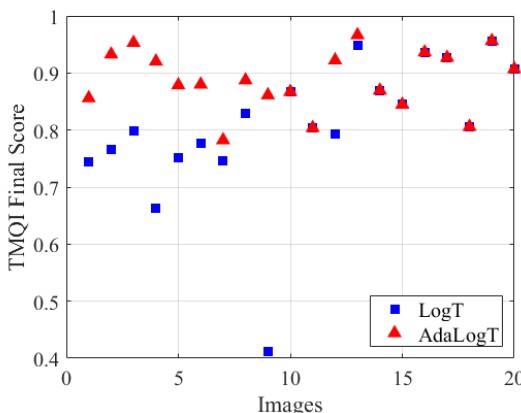

**Figure 11.** TMQI final score of the gradient-domain algorithm in different preprocessing methods.

**Table 3.** Mean TMQI scores of gradient-domain algorithm.

| Preprocessing | Structure | Naturalness | Final |
|---|---|---|---|
| LogT | 0.8104 | 0.2577 | 0.8072 |
| AdaLogT | 0.8647 | 0.5272 | 0.8879 |

### 5.3. DNN-Based TM Algorithm

Vinker [21] achieved good results by building a generative adversarial network to perform unpaired data training. Compared with its proposed adaptive curve-based compression (ACC), the objective function for the DNN-Based TM algorithm comprehensively considers image luminance and gradient characteristics, improving the subjective and objective quality of TM results. ACC uses an average of 900 image histograms as the primary target for parameter selection. The image exposure, skewness, and gradient priors are used in this paper to make the results have the same or even better display effect, as shown in Figure 12. The subjective scores of Vinker and our method are (7.11, 1.16) and (7.37, 1.06). According to the quantitative subjective evaluation, we have made certain progress in the subjective effect.

Figure 13 shows the TMQI quality scores of the two preprocessing methods for each experimental image, and Table 4 shows the mean TMQI scores of the TM algorithm [21] under different preprocessing methods. For the DNN-Based algorithm with complex features, the proposed method outperforms the ACC method in most images. Moreover, the enhancement of image gradients ensures that the TM algorithm results have better structure-preserving properties, achieving higher TMQI structure and naturalness scores.

**Table 4.** Mean TMQI scores of DNN-based TM algorithm.

| Preprocessing | Structure | Naturalness | Final |
|---|---|---|---|
| ACC | 0.8587 | 0.5398 | 0.8872 |
| AdaLogT | 0.8798 | 0.6383 | 0.9129 |

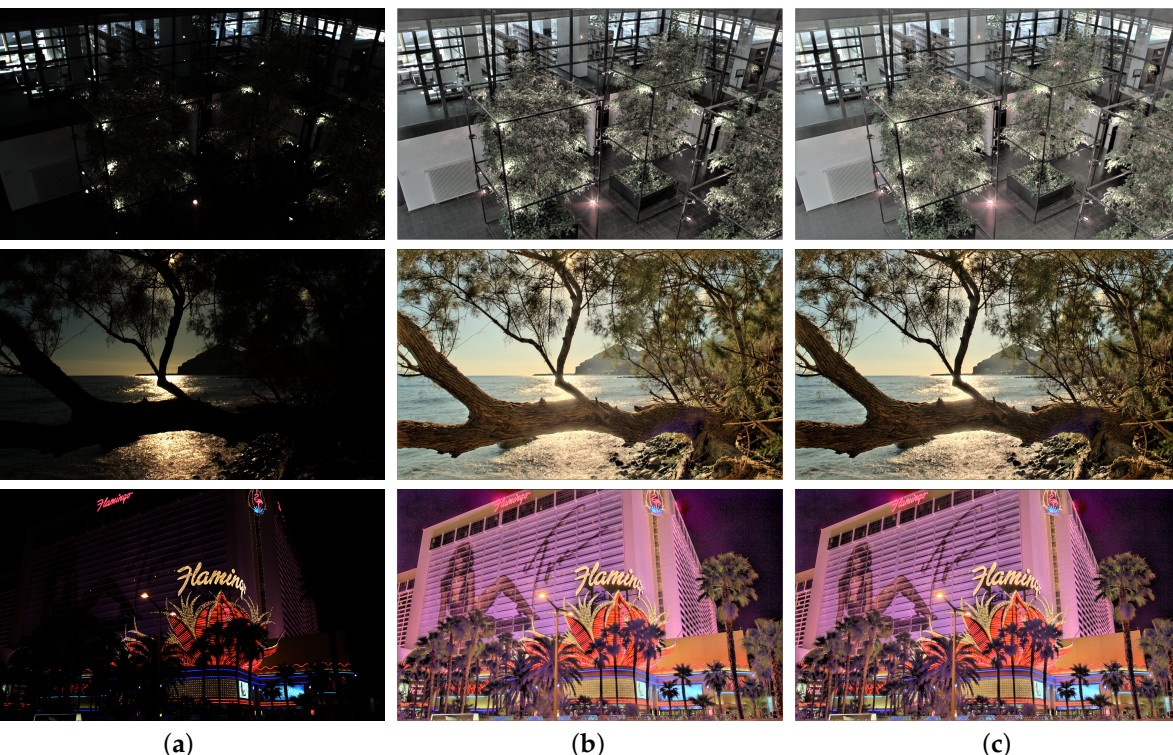

**Figure 12.** Comparison of the results of the DNN-based TM algorithm in different preprocessing methods: (**a**) radiance map; (**b**) ACC [21]; (**c**) AdaLogT.

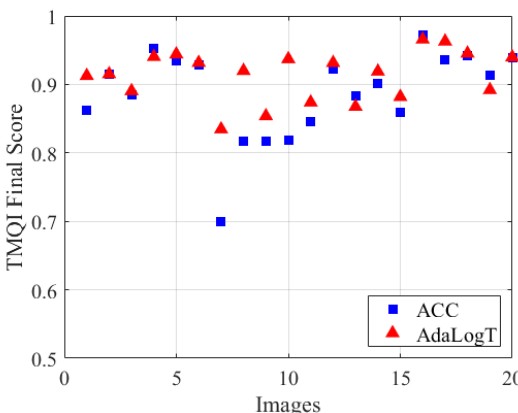

**Figure 13.** TMQI final score of the DNN-based TM algorithm in different preprocessing methods.

## 6. Conclusions

This paper proposed an adaptive logarithmic normalization transformation, AdaLogT, for TM algorithms in order to compensate for the defects caused by the preprocessing. Based on the analysis of classical preprocessing methods, the parameter $p$ was introduced in order to obtain the appropriate log-normalized curve. The optimal parameter $p$ was calculated by proposed objective functions. Considering the TM algorithms based on luminance or gradient domain, the objective functions based on luminance and gradient-domain features were constructed, respectively. Furthermore, a joint domain-aware objective function was presented for DNN-based TM algorithms. The proposed preprocessing algorithm ensures that the image luminance conforms to the HVS perception of the scene brightness. State-of-the-art luminance, gradient-domain, and DNN-based algorithms were selected for the experiments, which used different preprocessing methods. The experiments were conducted by combining subjective qualitative and objective quantitative. The results show

that the proposed algorithm achieves the best subjective quantitative scores with TMQI quality scores, which indicates that the method improves the subjective effect and objective quality of the images. Further work includes the improvement of the optimization method for optimal parameter $p$ and the utilization of folded concave functions for more general tone mapping preprocessing.

**Author Contributions:** Conceptualization, X.F. (Xuelai Fang) and X.F. (Xiangchu Feng); Supervision, X.F. (Xiangchu Feng); Writing—original draft, X.F. (Xuelai Fang); Writing—review and editing, X.F. (Xuelai Fang) and X.F. (Xiangchu Feng). All authors have read and agreed to the published version of the manuscript.

**Funding:** This work was funded by the National Nature Science Foundation of China under Grant 61772389 and Grant 61972264.

**Data Availability Statement:** All experimental data are from publicly available data sets. The URL for the datasets are: https://ivc.uwaterloo.ca/database/TMQI/TMQI-Database.html, https://pfstools.sourceforge.net/hdr_gallery.html, and https://qualinet.github.io/databases/image/tone-mapped-image-quality-database/(accessed on 21 October 2022).

**Acknowledgments:** We would like to express our sincere gratitude to the three reviewers for their insightful and valuable feedback, which has helped us improve our work.

**Conflicts of Interest:** The authors declare no conflict of interest.

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
