# Peer review of "Domain-Aware Adaptive Logarithmic Transformation"

_electronics, doi:10.3390/electronics12061318_

Round 1
Reviewer 1 Report
The authors propose a novel joint domain-aware logarithmic pre-processing method for deep-neural-network-based TM. Some comments are highlighted herewith for the improvisation of the manuscript.
1. Please incorporate the quantitative result in the abstract with regard to the proposed method.
2. Overall flowchart of the proposed method is missing in the methodology section.
3. Can the authors incorporate a table highlighting the pros and cons of the previous such studies done by the researchers.
4. Can the authors highlight the execution timing to achieve the desired result for a single image.
5. Please incorporate the specs of the machine on which the results were obtained.
6. Conclusion Section needs to be arranged properly.
Reviewer 2 Report
In my opinion, the main value of the paper is to compare different log-transformations for tone mapping of images. As a moderate contribution, the input luminescence scaling for one of the existing transformations is optimized. Please consider the following revisions.
1. Please do not repeat the same things multiple times.
2. Comparison examples of images should be all moved to Results sections, and some of those even to supplementary. In addition, the comparisons should be done more objectively, not just leave it up to the reader to decide.
3. All key terms and symbols must be defined, as many readers may not directly familiar with this research area.
4. The flow of ideas is not very clear, there are different metrics for image processing considered, but the purpose of doing so is unclear as the paper should be about log pre-processing for TM.
More specific comments:
- Abstract: define TM (purpose of), explain meaning of a domain, avoid using mathematical expressions, what is the difference between TM and TM algorithm - is it really algorithm or just an expression to calculate? What is objective function - why it is defined?
- does the base of the logarithm matter?
- Introduction should not contain Figure 2; move it to results
- Section 2: the ranges of values should be given for both inputs and outputs; l. 104: why 10^-4? what is logarithmic normalization transformation? It is unclear how Lambda should be computed from (6), perhaps argmin? What is the meaning of LDR in argument of H(LDR)?
- Section 3: auxiliary objective function is too generic, it needs to be properly, perhaps even formally defined, does it come from some optimization? what kind of optimization? l. 175: E not defined, in (12), it is unusual to use p_1 rather than p^ast to denote optimum, explain 'simple dichotomy method', methods discussed in Section 3.2 are not well defined
- (16) and (17) seem to have been introduced already in Section 2
- l. 239: EMD not defined
Reviewer 3 Report
This paper proposes a domain-aware adaptive logarithmic transformation method.
The paper is well written and the methodology and results are somewhat convincing.
However, there are some points that need to be addressed in the revision:
In Eqs. 1-11, the symbols "bar" and "tilde" are mixed and some of them are misused.
The terms "AdaLogT" and "AdaLog" are mixed.
In line 246, please elaborate the reason why Eq. 19 takes a greatly-reduced computational cost than Eq. 17.
In Fig. 11, what does "LogT" mean? The term has not been explained before.
The TMQI scores are provided for objective evaluation of the results. But there is nothing provided for subjective evaluation.
To show that the proposed method produces visually (or perceptually) better images, it is necessary to add some user study results.
